# Molecular Impacts of Dietary Exposure to Nanoplastics Combined or Not with Arsenic in the Caribbean Mangrove Oysters (*Isognomon alatus*)

**DOI:** 10.3390/nano11051151

**Published:** 2021-04-28

**Authors:** Marc Lebordais, Zélie Venel, Julien Gigault, Valerie S. Langlois, Magalie Baudrimont

**Affiliations:** 1Université de Bordeaux, CNRS, UMR EPOC 5805, Place du Dr Peyneau, 33120 Arcachon, France; Marc.Lebordais@ete.inrs.ca (M.L.); zelie.venel@gmail.com (Z.V.); 2Centre Eau Terre Environnement, Institut National de la Recherche Scientifique (INRS), 490 rue de la Couronne, Québec City, QC G1K 9A9, Canada; Valerie.Langlois@inrs.ca; 3Université Laval, UMI Takuvik 3376, 1045 Avenue de la Médecine, Québec City, QC G1V 0A6, Canada; julien.gigault@takuvik.ulaval.ca

**Keywords:** nanoplastics, bivalve, gene expression, trophic pathway, metal

## Abstract

Nanoplastics (NPs) are anthropogenic contaminants that raise concern, as they cross biological barriers. Metals’ adsorption on NPs’ surface also carries ecotoxicological risks to aquatic organisms. This study focuses on the impacts of three distinct NPs on the Caribbean oyster *Isognomon alatus* through dietary exposure. As such, marine microalgae *Tisochrysis lutea* were exposed to environmentally weathered mixed NPs from Guadeloupe (NPG), crushed pristine polystyrene nanoparticles (PSC), and carboxylated polystyrene nanoparticles of latex (PSL). Oysters were fed with NP-*T. lutea* at 10 and 100 µg L^−1^, concentrations considered environmentally relevant, combined or not with 1 mg L^−1^ pentoxide arsenic (As) in water. We investigated key gene expression in *I. alatus*’ gills and visceral mass. NP treatments revealed significant induction of *cat* and *sod1* in gills and *gapdh* and *sod1* in visceral mass. As treatment significantly induced *sod1* expression in gills, but once combined with any of the NPs at both concentrations, basal mRNA levels were observed. Similarly, PSL treatment at 100 µg L^−1^ that significantly induced *cat* expression in gills or *sod1* in visceral mass showed repressed mRNA levels when combined with As (reduction of 2222% and 34%, respectively, compared to the control). This study suggested a protective effect of the interaction between NPs and As, possibly by decreasing both contaminants’ surface reactivity.

## 1. Introduction

Plastic contamination is a worldwide environmental issue, as recognized by the United Nations Environment Program [1]. Daily used plastic products can last in the environment for several hundred years without being completely degraded [2,3]. With an average of 8 million tons of plastic waste discarded into oceans each year, Wright and Kelly (2017) estimated an accumulation of 250 million tons by 2025 [4]. Aquatic ecosystems act as natural receptacles for contaminants, including plastic debris, that are eventually carried by currents and converge to five main ocean areas called gyres [5]. Guadeloupe island is an example of a terrestrial area exposed to the North Atlantic gyre (seventh expedition continent in 2015, [6]). In the gyre, different sizes of plastic debris can be identified: large microplastics (size < 5 mm; [7]), microplastics (size < 0.3 mm; [8]), and nanoplastics (size < 0.001 mm; [9]).

Nanoplastics (NPs) are commonly referred to as plastic nanoparticles smaller than 1000 nm in one of the three dimensions of space [10]. NPs originating from industrial synthesis are referred to as primary NPs, whereas those from environmental degradation are called secondary NPs [8,11]. Indeed, nanofragmentation of plastic occurs within aquatic ecosystems through physical abrasions and chemical oxidations (e.g., waves, salt, and mainly UV; [12,13]), thus yielding more available additives used for specific plastic properties (e.g., plasticizers, UV-filters, flame retardants, and metals; [4,9]). As plastic debris age and fraction into NPs, more polar surfaces are formed, allowing contaminants to desorb or be adsorbed onto NP surfaces [4,14,15]. For example, Davranche et al. (2019) recently documented metallic adsorption on marine NPs [16]. Further analyses revealed that arsenic was one of the most abundant metals adsorbed on the collected plastics from Guadeloupean beaches [17]. Therefore, since most marine species ingest plastic debris regularly [18,19], NPs’ potential toxicity triggered questioning. Indeed, NPs have high residence time, surface reactivity, and size availability for cellular uptake [20,21] that present a great risk of interaction and bioaccumulation within organisms [22,23,24]. As such, nanoparticles have been classified as emerging contaminants [25]. However, there is a critical lack of data on NPs’ fate and bioavailability, mostly explained by quantification challenges [26,27,28]. For example, due to higher ionic strength in seawaters, it causes NP aggregation [26,27,28,29,30,31]. Yet Gigault et al. (2018b) newly observed limited aggregation of NPs passing through a salinity gradient [12], thus making ecotoxicological NP studies all the more relevant in estuarine areas such as mangroves. As a matter of fact, mangroves hold a significant role in ecosystem services [32]; also, their sensitive functioning provides helpful indicators to study worldwide issues [33,34].

To tackle the impacts of ingested NPs, we studied NPs’ effects alone and combined with arsenic (As) on Caribbean oysters. We conducted our experiment on *Isognomon alatus* wild oysters native to Guadeloupean mangrove swamps, since oysters are relevant organisms for ecotoxicological studies. Bivalves have indeed been commonly used as bioindicators for decades [35,36,37], particularly as monitoring organisms, given that they tolerate and accumulate high metal concentrations [32,38] and nanoparticles [39]. Additionally, plastic exposure of bivalve aquacultures recently raised concerns with regard to their putative role of NPs’ transfer into the food web up to humans [4,40,41]. Trophic exposure is an under-rated NPs pathway [42,43,44] that we explored through NPs-contaminated phytoplankton used to feed *I. alatus* oysters. Indeed, filter-feeding bivalves rely on ciliated structures to collect phytoplankton but also organic and inorganic particles (up to 500 nm; [45,46]). Thus, to mimic the environmental feeding conditions of *I. alatus*, solutions of the marine microalgae *Tisochrysis lutea* were used to expose oysters to three different NPs. Selected NPs for this study encompassed a custom of mixed Guadeloupean NPs to represent the environmental plastic weathering (NPG; secondary NPs), the crushed pristine polystyrene nanoparticles (PSC, primary NPs; [47]), and the additives-free synthetic carboxylated polystyrene nanoparticles of latex (PSL; [48]). Most NP studies have used monodispersed calibrated nanospheres, but these spherical NPs have been considered poorly relevant to address ecotoxicology issues [13,49]. Therefore, in the present study, we aimed to investigate gene expression changes of *I. alatus* exposed by the dietary route to NPs (10 and 100 µg L^−1^), in the presence or absence of As in the water. These NP concentrations are environmentally representative [50,51] and belong to the lowest range of in-laboratory NP exposures [49]. Our set of genes was chosen in order to target NPs and/or As mechanisms of action on endocytosis, cell cycle regulation, oxidative stress, mitochondrial metabolism, and detoxification, as those cellular functions are known to be commonly affected by nanoparticles and/or metallic contaminants [39]. Considering the ecotoxicological relevance of NPG as a mix of polydispersed NPs (with heterogeneous shapes, specific surface area, and oxidative degree), we also investigated the NPG’s putative role towards arsenic bioaccumulation into *I. alatus* gills and visceral mass. Overall, the main novelty of our study lies in comparing the effects of three relevant NPs, combined or not with arsenic, on *I. alatus* key genes.

## 2. Materials and Methods

### 2.1. Nanoplastic Dispersions

Plastics used to generate the mixed nanoplastics from Guadeloupe (NPG) were collected in 2016 on the Guadeloupean beaches (16°21′06″ N; 61°23′09″ W), naturally aged in situ by environmental factors (salt, mechanical abrasion, and mainly UV exposure). Polystyrene pristine pellets used to produce crushed nanoparticles (PSC) were commercially purchased from Goodfellow (Lille, France). Both nanoplastic dispersions were prepared according to an optimized protocol [47]. Plastics pellets were degraded with 99% ethanol in a blade grinder to get a primary powder and later fragmented using a planetary ball mill [47]. The resultant powder was then dried by lyophilization to remove ethanol, then suspended in ultra-pure water and filtered on cellulose acetate filters (5–6 µm pore size; VWR, Biare, France). Hydrodynamic diameters of NPG and PSC nanoparticle dispersions of 50 ppm (in total organic carbon) were measured by dynamic light scattering (DLS) (Table 1) through a contactless (in situ) DLS probe at 170 °C on a Vasco flex instrument (Cordouan Technologies ^®^, Pessac, France). The intensity fluctuations as a function of the time were processed as an autocorrelation function. The cumulants algorithm was used to fit this function in order to obtain a size distribution (z-average). The surface charges of particles (potential ζ) were assessed using a Wallis zetameter (Cordouan Technologies ^®^, Pessac, France) (Table 1). Details of NPG and PSC dispersions are, respectively, presented in Appendix A. The carboxylated polystyrene nanoparticles of latex (PSL) were synthesized and calibrated by IPREM (Pau, France) with 43 COOH groups per nm² and a z-average of 390 ± 20 nm. Additionally, they were made spherical with a raspberry-like surface texture [48].

Noteworthily, polystyrene was chosen for both PSL and PSC, as it is one of five main plastics produced, representing 90% of global demand [52]. All three nanoplastic particle sizes were suspended and diluted as needed in ultra-pure water (18 MΩ cm^−1^, Millipore water purification system, Merck, Darmstadt, Germany) in cleaned glass vials. The mass concentration of each nanoplastic dispersion was measured by total organic carbon (TOC) analysis on a Shimadzu ^®^ instrument (Europa GmbH, Duisburg, Germany).

### 2.2. Microalgae and Oyster Cultures

The microalgae species used to feed the oysters during the acclimation period were *Tisochrysis lutea* [41,53] and *Thalassiosira weissflogii* [54] obtained from Lycée polyvalent de la Mer (Gujan-Mestras, France). Of note, *T. weissflogii* was only used during oyster acclimation for nutritional purposes [55]. Both algal cultures were cultivated into glass balloons in F/2 medium at salinity 26‰ and oxygenated by an air distribution pump through a glass pipette. As recommended by Helm and Bourne (2004) [56], the microalgae were grown at 22 °C under 24:24 artificial light of 100 µmol/m²/s.

This study used native tropical flat oysters, as they are representative of mangrove swamps [57,58]. Individuals of *I. alatus* were collected from their natural habitat in Grand Cul-de-sac Marin (16°18′58.1460″ N; 61°32′1.9379″ O), a natural reserve to the north of Pointe-à-Pitre in Guadeloupe. Once brought to the laboratory, they were individually brushed to remove external parasites and stored in 30 L tanks (100 oysters per tank) lined with tiles to fulfill *I. alatus’* need for a hanging substrate. The tanks were filled with reconstituted seawater (Instant Ocean ^®^) at 32‰ salinity, oxygenated, and filtered by an aquarium filter pump. Additionally, 2/3 of the water tanks were renewed every two days during the first week of oysters’ acclimation given their high organic matter release. Similar to *I. alatus* native environmental conditions, they were acclimatized at 26 °C with aquarium heaters and under 12:12 (natural light: dark cycles) for 15 days according to A. Arini (personal communication, 2 February 2019). Oysters were fed twice a week with 100 mL of mixed *T. lutea* and *T. weissiflogii* (10 × 10^6^ cells L^−1^ and 2 × 10^6^ cells L^−1^, respectively) per tank.

### 2.3. Experimental Design of Trophic Exposure

A preliminary experiment was conducted to determine *T. lutea* optimal exposure to NP solutions (Figure 1). *Tisochrysis lutea* solutions were exposed in 10 mL individual glass vials with five replicates. Of note, all the glassware exposed to NPs was previously cleaned using an acid bath of 3% nitric acid, rinsed with distilled water, then with 70% ethanol, and dried under a fume hood. Cell concentrations were measured daily for 96 h by spectrophotometry at 750 nm. Based on microalgae concentration variability, we estimated that 48 h was the optimal duration to conduct the trophic exposure. Thus, NPG, PSC, and PSL were separately added into algal solutions for 48 h at low nominal concentrations of 10 and 100 µg L^−1^ reported to be relevant proxies of NPs’ environmental concentration [50,51]. Prior to each NP inoculation, the microalgae concentrations were controlled by counting on a Nageotte chamber and brought to 1 × 10^6^ cells mL^−1^. Each NP-*T. lutea* solution was exposed to similar abiotic conditions (media, temperature, light) as the acclimation phase previously mentioned. To avoid NP accumulation onto the air pump tubing, the NP-*T. lutea* solutions were oxygenated through an agitation table [6].

A single solution of dissolved pentoxide arsenic (hereafter referred to As) was used (Merck KGaA ^®^, Darmstadt, Germany). Among inorganic forms of arsenic, pentoxide arsenic has been chosen, as it is the most abundant form found in oxygenated marine waters [59,60,61,62]. Oyster exposure to 1 mg L^−1^ As was conducted through batch injection at the beginning of the exposure (day 0). This As nominal concentration was based on oyster bioaccumulation tolerance and As levels in seawater [63,64,65]. To keep the concentration consistent at 1 mg L^−1^ throughout the exposure, water samples were collected in the morning and As levels were quantified. Thus, As concentration was adjusted one day out of two if needed.

The experimental design encompassed (Figure 2) a control of reconstituted seawater, an As treatment of 1 mg L^−1^, three single-NP treatments (NPG, PSC, and PSL) at 10 and 100 µg L^−1^, and a mix of As (1 mg L^−1^) with each of the NP treatments (As + 10 µg L^−1^ NP and As + 100 µg L^−1^ NP). All treatments were run in quadruplicate with two oysters per 500 mL glass jar.

Each jar was parafilm-covered to minimize evaporative loss. Water oxygenation was ensured by an air distribution pump. The oysters were under the same abiotic conditions (salinity, temperature, and light) as the above-mentioned acclimation period. Oysters of each condition were fed every two days with 35 × 10^3^ cells oyster^−1^ L^−1^ microalgae considered as a relevant environmental concentration [66]. After one week of dietary exposure, individual oysters were assessed for biometric parameters (Appendix A). Each tissue was manually dried with paper, then weighed (fresh weight). Gills and visceral mass were sampled for arsenic bioaccumulation and molecular assays. Empty shells were also individually measured and weighed. Thus, the condition index [67] was calculated as follows in Equation.
(1)$$\begin{array}{l} \mathrm{CI}=\frac{\mathrm{leftover}\;\mathrm{tissues}\;*\;\mathrm{weight}}{\mathrm{shells}\;\mathrm{weight}}\times 100\\ {}*\mathrm{leftover}\;\mathrm{tissues}:\mathrm{whole}\;\mathrm{body}\;\mathrm{excluding}\;\mathrm{gills}\;\mathrm{and}\;\mathrm{visceral}\;\mathrm{mass} \end{array}$$

### 2.4. Water and Tissue Arsenic Quantification by ICP-OES

To evaluate the efficiency of the As dissolution and dilution, a nominal solution of 1 mg L^−1^ was prepared from a 1 g L^−1^ As solution. The 1 mg L^−1^ solution was then acidified with 3% nitric acid to measure the final concentration on an inductively coupled plasma optical emission spectrometry (ICP-OES 700 series, Agilent, Santa Clara, CA, USA). To monitor and compensate for the As concentration throughout the exposure week, water samples of 0.5 mL were collected per jar and acidified at 3% nitric acid for ICP-OES analysis. 

For tissue analyses, the pooled gills and visceral mass were dried (48 h at 50 °C) and weighted prior to digestion (dry weight). Tissue samples were then acidified with 70% nitric acid (3 mL per sample) and heated at 100 °C for 3 h on a hot plate (Digiprep, SCPScience, Québec, Canada). After dilution of the digestates with 18 mL of pure water, total As bioaccumulation was measured using an ICP-OES (Figure 3). Certified biological reference materials (dolt-5, [68]) were systematically analyzed with the samples to ensure that the data obtained were within the certified range.

### 2.5. RNA Extraction and cDNA Synthesis

Gill and visceral mass tissues were preserved in RNA later at −20 °C right after dissections. For minimal weight purposes, tissues from two oysters were pooled together. Tissues were ground with a Biorad Fastprep ^®^ (40 s at 6 movements/s) in 500 µL of RNA lysis buffer, using ceramic pellets (Lysing Matrix D Bulk from MP Biomedicals). Deproteinization was done by adding 500 µL of phenol-chloroform isoamyl alcohol, as this organic solvent is very suitable for oyster tissues rich in proteins and fat [69]. Then, samples were vortexed and centrifuged for 5 min at 9000× *g* to separate the organic and the aqueous phases. From there, total RNA was extracted using the Promega kit “SV Total RNA Isolation System” according to the manufacturer. Extracted RNA concentration was measured by using a microplate spectrophotometer at 260 nm, and RNA quality was estimated with a Take3 plate (nucleic acids purity ensured by 260/280 ratio ≥ 2), both from BioTek EPOCH ^®^. All samples were diluted to obtain the same RNA concentration (1000 ng in 10 µL) before performing reverse transcription to synthesize complementary DNA (cDNA) with the Promega kit “GoScript Reverse Transcription System” according to the supplier’s instructions.

Given the non-sequenced genome of *I. alatus*, genes of interest were selected from the reconstituted *I. alatus* transcriptome sequenced by Lemer (2019) [70]. For each gene, specific primer sets were designed using Primer3Plus. The primer details can be found in Appendix A. Our genes of interest were representative of five biological functions. Transport-pathway-related genes included *cav* (cell membrane invagination for endocytosis) and *cltc* (intracellular vesicle transport; [71,72]). Genes related to cell cycle regulation included *p53* (tumor suppressor; [73]), *gadd45* (DNA repair; [74]), and *bax* (apoptotic activator; [73]). Oxidative stress-related genes included *sod1* (dismutation of superoxides into O_2_ and H_2_O_2_), *cat* (conversion of H_2_O_2_ into O_2_ and H_2_O; [75]), and *gapdh* (reduction and oxidation activities; [76,77]). Mitochondrial metabolism-related genes included *cox1* (respiratory chain electron transport), normalized by *12S* being indicative of the ribosomic RNA level [78,79], and detoxification related genes included *mdr* (drugs’ cell expulsion; [80]).

### 2.6. qPCR Assays and Validations

A real-time quantitative polymerase chain reaction (qPCR) was performed with the Promega kit “GoTaq ^®^ qPCR Master Mix” containing the 5× buffer, the Taq polymerase, MgCl_2_, dNTP, and SybrGreen dye. Validation of the primer’s efficiency and specificity was conducted so that 1 µL of the forward and reverse primers (100 µM each) was optimal for conducting the reactions. Together with the qPCR mix, primers were added to 1/10 diluted cDNA. The resulting final volume was poured into white 96-well qPCR plates, including two no-template control wells without cDNA, and replaced by RNAse free-water. Plates were sealed and quickly centrifuged, then analyzed by the LightCycler 480 Roche ^®^ (Rotkreuz, Switzerland) starting with one cycle at 95 °C for 2 min, then 30 amplification cycles at 95 °C for 30 s and 60 °C for 30 s.

The amplification efficiency of qPCR primers was assessed by 10-fold sample serial dilutions. Specificity was determined for each reaction from the dissociation curve of the qPCR products, obtained by following the SyberGreen fluorescent level during gradual heating from 60 to 95 °C. Based on these indicators, non-conforming data were not analyzed (ND in Table 2). Following the 2^−ΔCt^ method described by Livak and Schmittgen (2001) [81], genes of interest were standardized by the average cycle thresholds (Ct) from the reference genes *β-actin* and *rpl7* [82]. Gills and visceral mass gene expression data are presented in Table 2.

### 2.7. Statistical Analysis

To satisfy the conditions for parametric analyses, raw data were transformed by commonly used functions (inverse square root or decimal logarithm). Normality was then confirmed using the Shapiro–Wilk test, and homoscedasticity was confirmed by the Student test (α = 0.05). Transformed data were then compared between each treatment and concentration exposure using a two-way analysis of variance (ANOVA). The significant differences were identified by a post-hoc Tukey HSD test on Prism 8.0 for gene expression and the Bonferroni test on SigmaPlot 12.0 for arsenic dosage.

## 3. Results

### 3.1. Oysters Biometric Parameters and Arsenic Bioaccumulation

Mortality during the exposure was negligible (<4%). No significant differences were observed for the total fresh tissue weights and the CI among treatments (data shown in Appendix A). Field-collected control animals had an average concentration of 75 µg g^−1^ As in gills and 133 µg g^−1^ As in visceral mass (Figure 3). As-exposed oysters yielded approximately 1.5 times greater As concentrations than the control oysters in both tissues (Figure 3). However, there were no statistical differences in As concentrations between the As and the NPG+As exposed oysters, nor between the two levels of NP exposure.

### 3.2. Relative Genes Expression in Gills and Visceral Mass

The expression of a suite of genes involved in five main biological functions, endocytosis (*cav* and *cltc*), cell cycle regulation (*p53*, *gadd45*, and *bax*), oxidative stress (*sod1*, *cat*, and *gapdh*), mitochondrial metabolism (*cox1* and *12S*), and detoxification (*mdr*) were assessed in gills and visceral mass (Table 2).

#### 3.2.1. Single Nanoplastic Treatments

In gills, the low exposure level of 10 µg L^−1^ to NPs revealed few genes with modulated expression, except repression of *gadd45* and *sod1* for PSC and PSL, as well as *cat* for PSL (Table 2). In contrast, for NP treatments at 100 µg L^−1^, NPG induced the expression of *cat* and PSL the expression of *sod1*, two genes implied in the response to oxidative stress. Additionally, PSC induced endocytosis by caveolin upregulation, while both PSC and PSL repressed *gadd45* expression, as observed for the lower concentration.

In visceral mass, the expression pattern of *gadd45* revealed a significant downregulation for both PSC and PSL at 10 µg L^−1^, but only for PSC at 100 µg L^−1^ (Table 2). At 10 µg L^−1^, PSC and PSL treatments revealed the induction of *gapdh*, which is implied in oxidative stress and further increased for PSL at 100 µg L^−1^. Moreover, the mRNA level of *sod1* was significantly upregulated for PSL at 100 µg L^−1^, in contrast to most of the other NP treatments for which *sod1* was repressed. The highest modulations of gene expression were observed for PSL at 100 µg L^−1^ by oxidative stress gene responses in addition to induction of *cltc* implied in endocytosis and *mdr* for detoxification mechanisms.

#### 3.2.2. Single Arsenic Treatment

In gills, As significantly induced the expression of *cox1*, revealing the mitochondrial metabolism impairment, and of *sod1*, revealing the oxidative stress burden. In parallel, *bax* was repressed.

In the visceral mass, interestingly, the expression of *cav* was significantly upregulated by the As treatment. In contrast, *bax* was significantly repressed. No statistical transcriptional changes were seen for *p53* expression between treatments; however, the ANOVA was significant for the test (*p* = 0.0009), which supports *p53* inhibition for As treatment.

#### 3.2.3. Nanoplastic Arsenic Treatments

In gills, the presence of NPs totally inhibited the induction of *sod1* and *cox1* previously observed with As alone, whatever the level of exposure or the type of NPs. The presence of As also canceled the induction of *cat* observed with NPG at 100 µg L^−1^ (return to the basal level of control) or even repressed its expression with PSL at 100 µg L^−1^ (reduction of 2222% compared to control).

In the visceral mass, the presence of NPs canceled the induction of *cav* observed with As alone, except for PSC+As at 100 µg L^−1^. In the same way, the induction of *cltc* observed for PSL at 100 µg L^−1^ was canceled in the presence of As. The expression of *bax* showed a significant downregulation for NPG+As at both 10 and 100 µg L^−1^, and only at 10 µg L^−1^ for PSC+As. Since PSL+As could not be detected, it is worth noting *bax* downregulation for PSL also at 10 µg L^−1^. The same downregulation pattern was shared in *gadd45* for PSC+As and PSL+As treatments at both concentrations. For *gapdh* expression, there was a significant downregulation for NPG+As at 10 µg L^−1^. The *sod1* expression response for PSL+As at 100 µg L^−1^ was significantly inhibited compared to the PSL treatment, and interestingly, aljso significantly lower than for the arsenic treatment. Expression of *mdr* was not modulated except for the NPG+As treatment, significantly inhibited at 10 and 100 µg L^−1^.

## 4. Discussion

### 4.1. Arsenic Uptake and Bioaccumulation in Oysters

The statistical analyses conducted on biometric parameters do not show any significant differences for the measured shell length and fresh tissue mass, nor for the calculated CI (Figure 3). Potential variations in As bioaccumulation in tissues can thus not be attributable to differences in biometric parameters.

The total As tissue bioaccumulation measured in controls demonstrated the biogeochemical background of Guadeloupean mangroves. These levels provide us with the environmental As baseline of *I. alatus* in their native habitat. Despite these rather high As values, they are comparable with a reviewed range of bioaccumulated As occurring naturally in bivalves [60,83]. For each treatment, total As bioaccumulation in gills is around half the As bioaccumulation in visceral mass. These results confirm the organ’s function towards chronic metal exposure with higher As levels in the visceral mass (storage organ; [84,85]) compared to the gills (transfer organ; [54,86]). This present study used 1 mg L^−1^ As, which allowed an increase in As bioaccumulation for both tissues after a short-term exposure (one week). No speciation analysis has been conducted, as we did not aim to address the As behavior in this study. Given chemical reactions in seawater, the initial As form (arsenate) underwent speciation changes [87,88,89]. Thus, it has to be kept in mind that total As results cannot be interpreted as the equivalent of the initial As form. Therefore, As treatment and total As results should be seen as a positive control. Seawater analysis from the oysters’ reference site was also conducted. For all sampling stations, total As concentrations measured were below the limit of detection. The chronic exposure of *I. alatus* is most likely responsible for the tissue bioaccumulation found in controls. Under environmental conditions, oyster exposure to As most likely came from sediment through the waterborne route [63,64,83] and phytoplankton through the dietary route, since microalgae bioaccumulate inorganic As forms [60,89,90]. Concentration has been only measured in NPG+As treatments, as NPG were the only nanoparticles carrying an initial As burden. Thus, NPG were the most indicated NPs to increase the As bioaccumulation in *I. alatus* tissues. Yet, there were no statistical differences in As bioaccumulation between the As and the NPG+As treatments, at 10 and 100 µg L^−1^ NPG concentrations for both tissues. Similarly, Freitas et al. (2018) did not observe an increased As bioaccumulation in clams in the presence of As (0.1 mg L^−1^) combined with multi-walled carbon nanotubes (0.1 mg L^−1^) [65]. Our results underlined that As bioaccumulation in both tissues was only driven by As treatment. Ultimately, the presence of NPs at these low concentrations did not affect the As total uptake of *I. alatus*.

### 4.2. Microalgae Growth under Nanoplastic Exposure

Phytoplankton organisms like microalgae are primary producers. They are therefore a keystone in aquatic food webs. Yet, NPs are known to interact with micrometric compounds such as microalgae due to their colloidal behavior in water [12,91,92]. To anticipate NP effects on *T. lutea* growth, a 96 h exposure was conducted with NPG, PSC, and PSL solutions at 10 and 100 µg L^−1^ (Figure 1). 

No significant changes were observed under any NP treatments. Noteworthily, higher optical density dispersions happened at 72 h for both NP concentrations. Thus, we decided to expose *T. lutea* for 48 h to NP solutions prior to feeding oysters. NPs’ availability in our trophic experiment is believed to be mainly driven by their filtration rate [39,93]. Therefore, special attention was given to feed oysters with consistent microalgae concentrations among treatments. NPs were expected to be adsorbed or internalized after 48 h exposure, and to potentially trigger physiological changes on the microalgae membrane. To assess the fate of NPG, PSC, and PSL at 10 and 100 µg L^−1^, optical microscopic observations were conducted on NP-*T. lutea*. We did not observe significant effects for any treatments. Yet, in a similar experiment (Lebordais et al., 2021) [94], we attested PSL adsorption on *T. lutea* surface for a concentration grade including 10 and 100 µg L^−1^ by scanning electron microscopy (SEM).

### 4.3. Effects of Nanoplastics and Arsenic on Genes Expression in Oysters

#### 4.3.1. Single Nanoplastic Treatments

Our results showed significant downregulation of the *gadd45* mRNA level for PSC and PSL treatments at both concentrations, except for PSL 100 µg L^−1^ in visceral mass (Table 2). Given that the main role of *gadd45* is to maintain genomic stability, this gene is strongly regulated by DNA damaging agents like metals, but also growth-arresting signals [74]. Oxidative stress has also been established by in vitro and in vivo studies as an early indicator of NP toxicity in freshwater bivalves [39,95]. Thus, NP toxicity is mainly exerted through two pathways: either directly by interacting with the cellular contents including DNA, or indirectly by ROS release, known for causing nucleotide adducts [96,97]. Significant upregulation for PSL treatment at 100 µg L^−1^ of *sod1* and *cat* in gills, as well as *sod1* and *gapdh* in visceral mass, might imply higher ROS production. The same scenario appears in gills for NPG treatment at 100 µg L^−1^ with the induction of *cat*, suggesting oxidative stress especially for NPG and PSL exposure conditions. Also for PSL at 100 µg L^−1^, the induction of *cltc* suggests a facilitated uptake of this kind of NPs in the cells. Yet, the detoxification process of *mdr* is also overexpressed and thus reveals the potential ability of cells to expulse these NPs. Our overall results thus showed different patterns of effects among NPs, thus shedding light on the necessity to use several NPs for ecotoxicological studies. Unlike conventional commercial nanoparticles [20,98,99], here we used functionalized PSL nanoparticles with no additives to avoid additional toxicity [48]. Indeed, most commonly used additives like sodium dodecyl sulfate, Tween ^®^, and Triton-X ^®^ can induce toxic effects on aquatic organisms [100,101,102]. Recently, it has been demonstrated by Pikuda et al., (2019) that the toxicity of commercial PSL-COOH on daphnia was coming from its bactericide additive (sodium azide) [103]. Additionally, since carboxylated functions ensured PSL stability, we propose its higher availability by endocytosis. It can be hypothesized that PSL was either more absorbed or adsorbed by microalgae before being ingested by oysters. Nanoparticles’ instability has indeed been acknowledged for turning aggregates into bigger particles (e.g., MPs), particularly in seawater solutions [10,29,31]. Natural Organic Matter has also been addressed to affect PS NPs aggregation, depending on the water chemistry and its ions valence [104]. Moreover, Mao et al. (2020) observed less aggregation from PS NPs in the presence of algal extracellular polymeric substances [31]. Interestingly, they also observed less aggregation from artificially UV-aged PS NPs, since the formation of carbonyl groups enabled their stabilization. In our experiment, lower stability has been particularly observed for NPG compared to the other NPs. Therefore, the aggregation of NPG may have led to less interaction with microalgae during the exposure. This could explain its fewer toxicity effects compared to PSC and PSL in visceral mass. As such, our results underlined the value to assess NPs’ stability in the exposure media [26,51] and to further study comparative NP effects. This is also supported by Baudrimont et al.’s (2020) study [6], which revealed a significantly higher production of pseudofeces by *Corbicula fluminea* after 36 h of exposure to 1000 µg L^−1^ environmental NPs, which was not observed under conventional commercial polyethylene NP exposure. These comparative NP exposures, along with our current results, suggest different hazardous effects between environmental and conventional plastic nanoparticles. Thereby, we would like to raise awareness of the underestimation of NP ecotoxicity in the literature, as most conventional commercial polystyrene nanoplastics are being used.

#### 4.3.2. Single Arsenic Treatment

Data showed *bax* to be downregulated following As treatments in both tissues. It has already been described in humans that As is able to inhibit the repair of DNA damages, leading to carcinogenic effects [105]. In gills, the significant overexpression of *sod1* and the upregulation of *cox1* should indicate in this tissue generation of oxidative stress and mitochondrial metabolism disruption, respectively. Our results indeed revealed a metal effect similar to that observed by *cox1* upregulation in *C. fluminea* hemocytes exposed to ionic Au for 28 days [39].

#### 4.3.3. NPs+As Treatments

Expression of *mdr* revealed opposite responses at 100 µg L^−1^ between PSL treatment’s upregulation and NPG+As treatment’s downregulation in visceral mass. MXR pumps work as a cytoprotective system by excreting exogenous compounds. Nonetheless, it is sensitive to both physical and chemical stressors [106]. NP toxicity might be increased by As interaction with NPG, since the detoxification process can be inhibited in the presence of metals [107]. Similar results were found for encoded MXR-related proteins in *Mytilus galloprovincilalis* digestive glands exposed to 0.75 µM Hg^2+^ for six days [80]. Due to physical and chemical fragmentation factors [11,108,109], NPG surfaces are heterogeneously degraded and expected to be the most environmentally representative NPs in aquatic ecosystems. As proposed by Lambert and Wagner (2016), Ter Halle et al. (2017), and Baudrimont et al. (2020) higher reactivity of secondary plastic surfaces enables increased availability for contaminants to be adsorbed onto them [6,9,28]. This experiment evaluated NP affinities towards As, showing more interaction for NPG and PSL (as expected from NPG degraded surfaces and PSL carboxylated surface groups). Previous studies recorded changes in bivalve closure rhythm under metal exposure, ultimately affecting their filtration [110,111]. This stress is most likely to be seen in the transcriptional response of gills, but also in visceral mass tissues. Similar bioavailability of As for *I. latus* most likely explains the converging responses between NPs+As treatments and single As treatment for cell cycle regulation (*bax*). Yet, we observed lower toxicity in NPs+As treatments compared to single As treatments for endocytosis (*cav*), oxidative stress (*sod1*), and mitochondrial metabolism (*cox1*). Antagonist effects have been recently documented for MP-NP exposures combined with metals [112,113] including As [66]. Antagonist effects refer to significantly lower responses in combined contaminant treatments (NPs+As) compared to the control and both individual contaminants [114]. However, we underline here basal mRNA levels for combined NPs+As treatments. We are not aware of any earlier studies showing a decrease of NPs’ gene expression effects when in the presence of any metal exposure. Therefore, we aim to repeat this experiment with another oyster species to confirm this novel finding.

### 4.4. Models of Trophic Exposure Effects on the Studied Gene Functions

Based on the genes mRNA levels, Figure 4 sheds light on the main findings of this present study. Induction arrows show the presence of a biological effect for a given treatment, regardless of genes’ up- or downregulation. Prevention arrows represent the absence of a biological effect for NPs+As treatment. Therefore, the gene response is equivalent to the control treatment. An antagonist arrow shows the presence of a biological effect for NPs+As treatment. Therefore, the gene response is significantly lower than the control and absent in NP treatments.

In gills (Figure 4A), PSC internalization can be hypothesized at 100 µg L^−1^ by *cav* induction. Since a fraction of PSC might be aggregated, resulting in bigger particles similar to MPs, a higher PSC concentration might be required to witness their internalization. Polystyrene nanoplastics (PS NPs) were consistent in DNA damage, showing no change of toxicity for PSC+As and PSL+As treatments on *gadd45* repression. Additionally, the change of *bax* expression for NPG+As treatment at 10 µg L^−1^ was most likely caused by As. Contrary to what was expected, no synergetic effects were observed between As and NPs in combined treatments. At a subcellular level, the As treatment alone affected mitochondria homeostasis by *cox1* upregulation, but once in presence of any NPs, *cox1* upregulation was canceled. Thus, we hypothesize that NPs’ presence could decrease the As availability, most likely through As adsorption on NPs, but no affirmation can be done, as there were no significant differences. In the presence of As, there was an inhibition of *cat* expression for PSL +As treatments at 10 and 100 µg L^−1^, contrary to NP single treatments. Moreover, As alone induced oxidative stress revealed by *sod1* expression and so did the PSL treatment at 100 µg L^−1^, but once As was in presence of PSL (PSL+As treatment), a putative decrease of oxidative stress was observed. Therefore, data highlight a potential protective role of PSL against the oxidative stress caused by As, but also a protective role of As toward the oxidative stress generated by NPG and PSL at 100 µg L^−1^ revealed by the *cat* upregulation.

In visceral mass (Figure 4B), data suggest PSC internalization in the presence of As at 100 µg L^−1^ by *cav* induction, leading to membrane invagination. PS NPs triggered DNA damage given the *gadd45* repression. Based on their physico-chemical characteristics, PSC and PSL yielded a low and very low endogenous ROS burden, respectively. Thus, DNA damage most likely came from the physical nanoparticle toxicity, excluding additional toxicity from As in PSC+As and PSL+As treatments. However, an opposite pattern was observed for NPG+As, where the *bax* mRNA level decreased once NPG were in presence of As. Therefore, As interaction showed different effects among NPs toward cell cycle regulation and apoptosis. Moreover, PSL and NPG expressed an opposite *mdr* response at 100 µg L^−1^. Indeed, as PSL were very stable, an increased exposure seemed to lead to a greater induction response from the cell to expel them. Oxidative stress was triggered by PSL at 100 µg L^−1^ inducing *sod1* regulation, but was canceled in presence of As (PSL+As treatment), while *gapdh* expression was inhibited by NPG+As treatment at 10 µg L^−1^. Thus, As presence seemed to cancel the oxidative stress production seen for NP single treatments. It has to be kept in mind that NPG were the most weathered NPs, with high porosity and high surface oxidative degree that potentially increased its interaction with As. The same logic applies for PSL by As adsorption on its carboxylated functions. Noteworthily, As treatment induced endocytosis activation by *cav* expression. Again, we may have observed As interaction with NPs through adsorption that led to canceling *cav* induction inNPG+As treatment at 10 µg L^−1^.

## 5. Conclusions

We demonstrated that sublethal exposures of NPs impaired cellular functions at the molecular level on native Guadeloupean oysters. Among our three NP solutions, PSL treatments showed the most consistent toxic effects by triggering oxidative stress and cell cycle regulation in gills and visceral mass tissues. These results shed light on the interest to use PSL additives-free nanoparticles with surface functionalized groups (e.g., carboxyl group) rather than conventional commercial PSL. Similarly, but with fewer effects observed, PSC toxicity came most likely from their irregular surface, with a highly specific surface that potentially enabled more adsorption on the microalgae surface. Interestingly, NPG treatments also showed toxic effects, but fewer than PSL by only triggering oxidative stress in gills. In light of these results, we suggest that studies might include more complex environmental NPs in their study design. Moreover, to our knowledge, we present the first results showing a specific response of NPs+As exposure with a protective effect on oxidative stress and on endocytosis. All these results underline the threat of plastics for marine wildlife and the need to study crossed effects between NPs and other contaminants.

## Figures and Tables

**Figure 1 nanomaterials-11-01151-f001:**
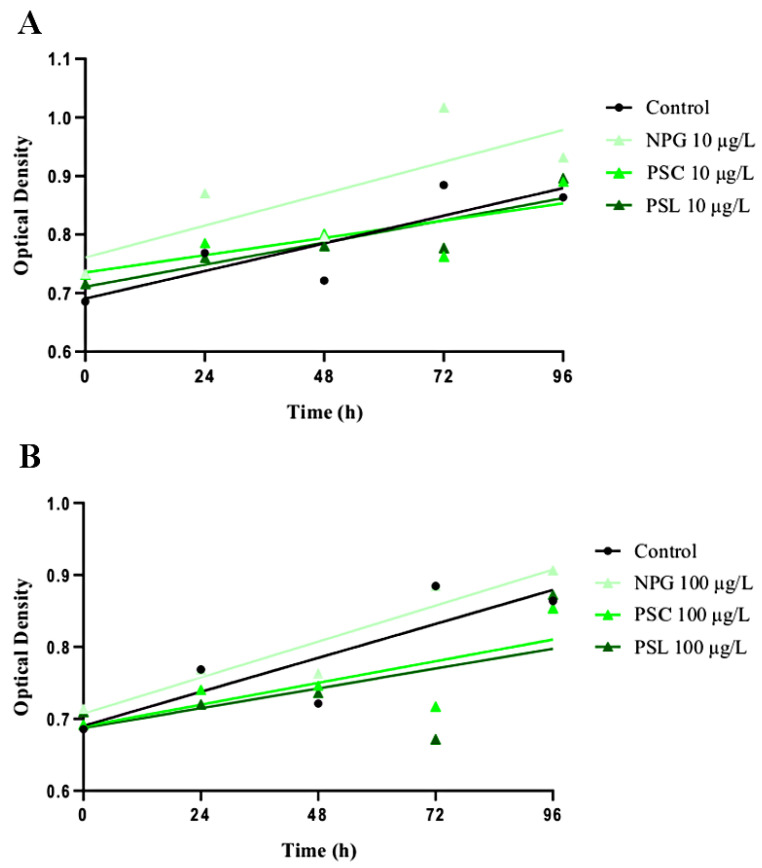
*Tisochrysis lutea* growth cultured in F/2 medium for 96 h. Linear regressions with mean points (*n* = 5) are represented for 10 µg L^−1^ NP treatments (**A**) and 100 µg L^−1^ NP treatments (**B**). Regression coefficients (**A**) are control = 0.74, NPG = 0.60, PSC = 0.61, PSL = 0.80. Regression coefficients (**B**) are control = 0.74, NPG = 0.90, PSC = 0.59, PSL = 0.33.

**Figure 2 nanomaterials-11-01151-f002:**
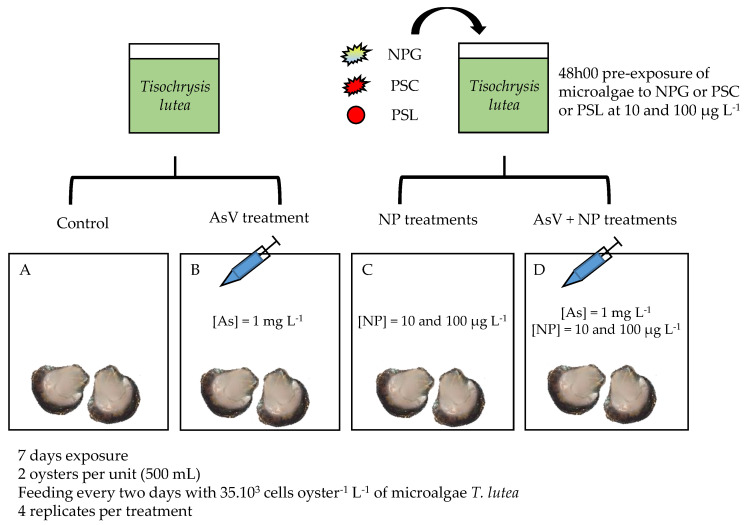
Experimental procedure of I. alatus trophic exposure. Condition (**A**) corresponds to control. Arsenic injections were performed at day 0 in treatments (**B**,**D**). NPs were added through pre-exposed microalgae feeding, one day out of two in treatments (**C**,**D**). NP refers to individual NPG, PSC, or PSL treatments.

**Figure 3 nanomaterials-11-01151-f003:**
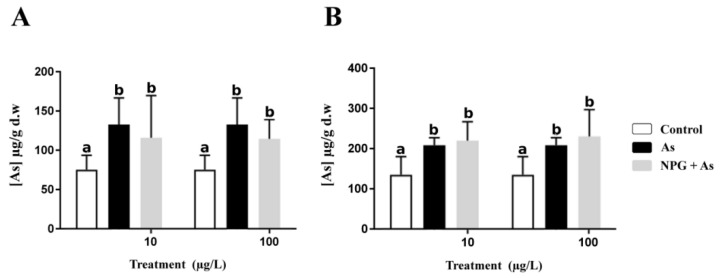
Arsenic accumulation in gills (**A**) and visceral mass (**B**) (µg g^−1^ dry weight, mean + sd), after one week of exposure to 0 (Control) and 1 mg L^−1^ of As alone or combined with NPG (10 and 100 µg L^−1^). Different letters denote statistical differences between treatments assessed by two-way ANOVA followed by Bonferonni post-hoc test (*n* = 4); *p* < 0.05.

**Figure 4 nanomaterials-11-01151-f004:**
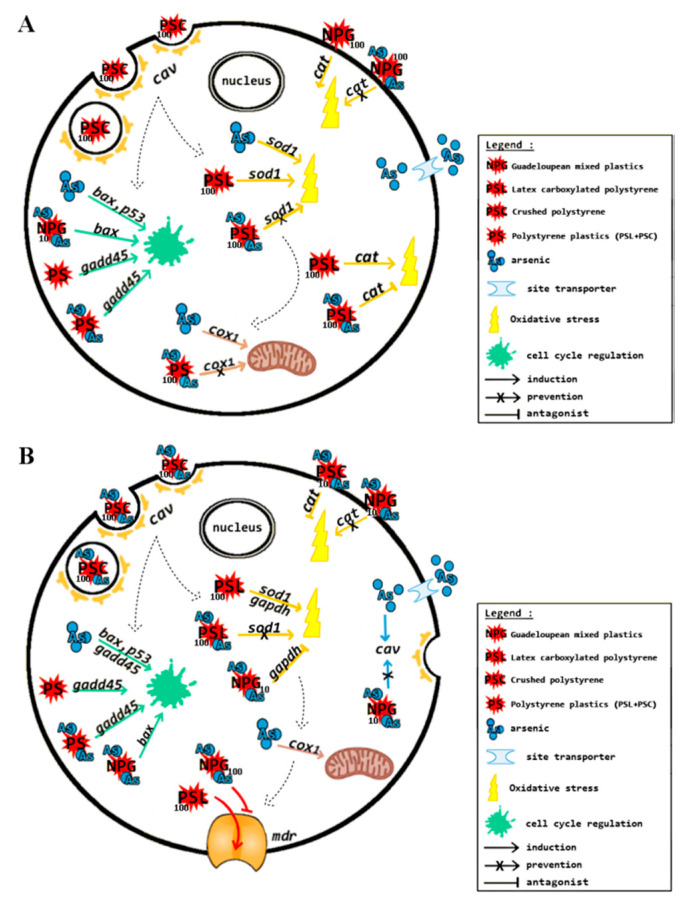
Schematics of suggested cellular effects of NPs and As in *I. alatus’* gills (**A**) and visceral mass (**B**) tissues.

**Table 1 nanomaterials-11-01151-t001:** Main characteristics of NP dispersions measured by DLS in situ on a Vasco Flex instrument (Cordouan Technologies ^®^, Pessac, France).

Nanoparticles	Z-Average (nm)/PDI	Potential ζ (mV) 5 mM NaCl, pH 7
NPG	361 ± 40 nm/0.210	−30.2 ± 1.1
PSC	354 ± 30 nm/0.190	−44.0 ± 2.0
PSL	390 ± 20 nm/0.002	−42.0 ± 2.0

**Table 2 nanomaterials-11-01151-t002:** Relative gene expression in *I. alatus’* gills (**A**) and visceral mass (**B**) after one week of exposure to 10 and 100 µg L^−1^ NPs by the dietary route, combined or not with 1 mg L^−1^ As in water. All the values are presented as the mean ± sd (*n* = 4) of fold changes compared to control (induction in green and repression in red), previously normalized by *β-actin* and *rpl7* reference genes. Different letters (*a* and *b*) denote statistically significant differences (*p* < 0.05) between treatments for each gene assessed by two-way ANOVA followed by Tukey post-hoc test. / indicates no statistically significant differences with the control.

*A—Gill.*														
*Functions*	*Genes*	*NPG* *10*	*PSC* *10*	*PSL* *10*	*NPG* *100*	*PSC* *100*	*PSL* *100*	*As*	*As+NPG* *10*	*As+PSC* *10*	*As+PSL* *10*	*As+NPG* *100*	*As+PSC* *100*	*As+PSL* *100*
*Endocytosis*	*cav*	/	/	/	0.14 ± 0.21*a*	2.78 ± 1.10*b*	/	/	/	/	/	/	/	/
	*cltc*	/	/	/	/	/	/	/	/	/	/	/	/	/
*Cell cycle regulation*	*bax*	/	ND	ND	/	ND	ND	0.03 ± 0.01*a*	0.28 ± 0.29*a*	ND	ND	/	ND	ND
	*gadd45*	/	0.20 ± 0.06*a*	0.14 ± 0.03*ab*	/	0.35 ± 0.14*a*	0.29 ± 0.11*ab*	/	/	0.11 ± 0.01*b*	0.17 ± 0.07*ab*	/	0.11 ± 0.04*b*	0.11 ± 0.01*b*
	*p53*	/	/	/	/	/	/	/	/	/	/	/	/	/
*Oxidative stress*	*cat*	/	/	0.08 ± 0.02*a*	7.13 ± 3.29*b*	ND	4.50 ± 1.54*b*	/	/	ND	0.07 ± 0.08*ac*	/	0.12 ± 0.04*a*	0.01 ± 0.00*c*
	*gapdh*	/	/	/	/	ND	/	/	/	/	/	/	/	/
	*sod1*	/	0.33 ± 0.05*a*	0.17 ± 0.07*a*	/	ND	7.89 ± 1.13*b*	2.86 ± 0.68*b*	/	0.30 ± 0.10*a*	/	/	/	/
*Detoxification*	*mdr*	/	/	/	/	ND	/	/	/	/	/	/	/	/
*Mitochondrial metabolism*	*12S*	/	/	/	/	0.43 ± 0.11*a*	/	/	/	/	0.24 ± 0.05*a*	ND	0.32 ± 0.12*a*	0.30 ± 0.11*a*
	*cox1*	/	/	/	/	/	/	7.33 ± 3.81*a*	/	/	/	/	/	0.22 ± 0.11*b*
***B—Visceral mass.***														
***Functions***	***Genes***	***NPG*** ***10***	***PSC*** ***10***	***PSL*** ***10***	***NPG*** ***100***	***PSC*** ***100***	***PSL*** ***100***	***As***	***As+NPG*** ***10***	***As+PSC*** ***10***	***As+PSL*** ***10***	***As+NPG*** ***100***	***As+PSC*** ***100***	***As+PSL*** ***100***
*Endocytosis*	*cav*	/	/	/	/	ND	/	33.74 ± 23.30*a*	0.13 ± 0.05*b*	/	/	/	15.62 ± 11.55*a*	/
	*cltc*	/	/	/	/	/	3.39 ± 1.32	/	/	/	/	/	/	/
*Cell cycle regulation*	*bax*	/	/	0.18 ± 0.04*a*	/	/	/	0.03 ± 0.01*b*	0.07 ± 0.06*ab*	0.18 ± 0.03*a*	ND	0.14 ± 0.03*a*	/	ND
	*gadd45*	/	0.21 ± 0.10*a*	0.24 ± 0.04*a*	/	0.18 ± 0.13*a*	/	/	/	0.18 ± 0.04*a*	0.18 ± 0.04*a*	/	0.21 ± 0.11*a*	0.23± 0.11*a*
	*p53*	/	/	/	/	/	/	/	/	/	/	/	/	/
*Oxidative stress*	*cat*	/	/	/	/	/	/	/	/	ND	ND	/	/	ND
	*gapdh*	/	1.66 ± 0.15*a*	1.35 ± 0.08*a*	/	/	2.89 ± 1.11*a*	/	0.30 ± 0.20*b*	/	/	/	/	1.81 ± 0.99*a*
	*sod1*	0.49 ± 0.29*a*	/	0.42 ± 0.28*a*	0.49 ± 0.41*a*	0.29 ± 0.03*a*	7.63 ± 3.18*b*	/	/	0.26 ± 0.05*a*	0.36 ± 0.14*a*	/	/	0.39 ± 0.26*a*
*Detoxification*	*mdr*	/	/	/	/	/	2.73 ± 0.59*a*	/	0.47 ± 0.36*b*	/	/	0.26 ± 0.10*b*	/	/
*Mitochondrial metabolism*	*12S*	/	/	/	/	/	/	/	/	/	/	/	/	/
	*cox1*	/	/	/	/	/	/	/	/	/	/	/	/	/

## Data Availability

Data available on request. The data presented in this study are available on request from the corresponding author.

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
