# Peer review of "Molecular Impacts of Dietary Exposure to Nanoplastics Combined or Not with Arsenic in the Caribbean Mangrove Oysters (Isognomon alatus)"

_nanomaterials, 2021, doi:10.3390/nano11051151_

Round 1

Reviewer 1 Report

The authors have made a great effort to construct a nice and sound paper about the impacts of nanoplastic, through food exposure, on As toxicity in mangrove oyster. This study can be be consider for publication in this journal its present shape.  

Author Response

Thank you very much for the nice comments made on our manuscript.

Reviewer 2 Report

The ms addressed the emerging challenge to assess the mixture toxicity of nanoscale plastic materials and inorganic pollutants known to cause an impact on marine species. The proposed prey-predator approach also is original and provides insight into real exposure scenarios and trophic transfer of contaminants + nanoplastics from phytoplancton to filter-feeders. The manuscript has been significantly improved in all its parts, both in the methodological details and in the integration of the results as indicated by the previous reviewers. The data are better presented and clearer and the discussion is articulated, providing value to the results obtained, underlining their novelty. Minor comments are listed below with the aim to improve its clarity and suitability to be published in the journal.

Line 27- Can you add a percentage of reduction upon combined exposure to As?

Line 81- …and perhaps to humans due to their consumption?

Line 109- which concentration of NP stock was used for their characterization (i.e. DLS) and which media has been used for suspensions?

Line 174- Figure 1 is missing in the ms, please add

Table 2 legend_ indicate in the legend the meaning of red and green colors. Check also decimal numbers and dots and commas. Finally indicate the meaning of letters “a” and “b”.

Lines 217-218 check “expression” (two times)

Line 328- check “responses”

Lines 343-347 check “cancelled” sound inappropriate, perhaps reduce? To what extent? Can you calculate a percentage of reduction or express this statistically?

Lines 363-365- Suggest to place this sentence in methods as a justification of the choice of the specific form of As used in the study.

Line 374 check the word “enabled”

Author Response

We want to thank the reviewer for his/her valuable comments on our manuscript. Here are our responses point by point to the requested corrections.

Line 27- Can you add a percentage of reduction upon combined exposure to As?

We added in the abstract the percentages of reduction of genes cat in gills and sod1 in visceral mass for PSL100 + As condition compared to controls. That is to say that the 100% of induction was considered as the response in case of PSL100 alone. So, for cat in gills, the reduction of gene expression was of 2222 % compared to control (100% = x4.5, so x0.01 = -2,222%) and for sod1, it was of 33.7% (100% = x7.6, so x0.39 = -33.7%).

Line 28. « (reduction of 2 222% and 34% respectively compared to control) »

Line 81- …and perhaps to humans due to their consumption?

Yes, absolutely, we totally agree with the reviewer, so we added Line 82 « …up to humans… »

Line 109- which concentration of NP stock was used for their characterization (i.e. DLS) and which media has been used for suspensions?

We added Line 148 that the suspensions were prepared in ultra-pure water, and Line 122 that the concentration of dispersions for DLS measurements was of 50 ppm of total organic carbon.

Line 174- Figure 1 is missing in the ms, please add

In fact, the figure 1 was inserted in the discussion section Line 407. We have moved this figure to Line 191.

Table 2 legend_ indicate in the legend the meaning of red and green colors. Check also decimal numbers and dots and commas. Finally indicate the meaning of letters “a” and “b”.

The legend of table 2 was completed with the signification of the green and red colors corresponding respectively to induction and repression factors (Line 320). The significativity of letters a and b was added (Line323). And the commas were corrected by dots in the entire table.

« Table 2. Relative genes expression in I. alatus’ gills (A) and visceral mass (B) after one week exposure to 10 and 100 µg L-1 NPs by dietary route, combined or not with 1 mg L-1 As in water. All the values are presented as the mean ± sd (n = 4) of fold changes compared to control (induction in green and repression in red), previously normalized by β-actin and rpl7 reference genes. Different letters (a and b) denote statistical significant differences (p < 0.05) between treatments for each gene assessed by two-way ANOVA followed by Tukey post-hoc test. / indicates no statistical significant differences with the control. »

Lines 217-218 check “expression” (two times)

I’m sorry but I didn’t find any repetition of the term « expression » in the entire manuscript.

Line 328- check “responses”

This was corrected Line 342 in the revised manuscript.

Lines 343-347 check “cancelled” sound inappropriate, perhaps reduce? To what extent? Can you calculate a percentage of reduction or express this statistically?

The term « cancelled » is used when the fold change of the gene expression returns to the basal expression level of the control (that is to say a fold change of 1). The « repression » is used when the fold change is inferior to 1 and statistically different from the control. Only the fold changes significantly different from controls are presented in Table 2. For those with no differences compared to control, it is mentionned as /.

So, to calculate a reduction percentage of genes expression between conditions of NPs alone and NPs+As, the 100% is considered as the induction obtained with NPs alone compared to control. So, the repression observed in comparison for PSL100+As is of 2 222% compared to control, as explained previously. This was added in the text Lines 353 to 356.

« The presence of As also cancelled the induction of cat observed with NPG at 100 µg L-1 (return to the basal level of control) or even repressed its expression with PSL at 100 µg L-1 (reduction of 2 222% compared to control). »

Lines 363-365- Suggest to place this sentence in methods as a justification of the choice of the specific form of As used in the study.

The sentence has been moved to the methods section, Lines 198-200.

Line 374 check the word “enabled”

The term « enabled » was replaced by « allowed » Line 389.